# Elucidating the Role of Wildtype and Variant FGFR2 Structural Dynamics in (Dys)Function and Disorder

**DOI:** 10.3390/ijms25084523

**Published:** 2024-04-20

**Authors:** Yiyang Lian, Dale Bodian, Amarda Shehu

**Affiliations:** 1School of Systems Biology, George Mason University, Manassas, VA 20110, USA; ylian@gmu.edu; 2Diamond Age Data Science, Boston, MA 02143, USA; dale@diamondage.com; 3Department of Computer Science, George Mason University, Fairfax, VA 22030, USA

**Keywords:** fibroblast growth factor receptor 2 (FGFR2), tyrosine kinase domain, missense mutations, constitutive activation, structural dynamics, regulatory functions, clinical implications

## Abstract

The fibroblast growth factor receptor 2 (*FGFR*2) gene is one of the most extensively studied genes with many known mutations implicated in several human disorders, including oncogenic ones. Most FGFR2 disease-associated gene mutations are missense mutations that result in constitutive activation of the FGFR2 protein and downstream molecular pathways. Many tertiary structures of the FGFR2 kinase domain are publicly available in the wildtype and mutated forms and in the inactive and activated state of the receptor. The current literature suggests a molecular brake inhibiting the ATP-binding A loop from adopting the activated state. Mutations relieve this brake, triggering allosteric changes between active and inactive states. However, the existing analysis relies on static structures and fails to account for the intrinsic structural dynamics. In this study, we utilize experimentally resolved structures of the FGFR2 tyrosine kinase domain and machine learning to capture the intrinsic structural dynamics, correlate it with functional regions and disease types, and enrich it with predicted structures of variants with currently no experimentally resolved structures. Our findings demonstrate the value of machine learning-enabled characterizations of structure dynamics in revealing the impact of mutations on (dys)function and disorder in FGFR2.

## 1. Introduction

Receptor tyrosine kinases (RTKs) are a crucial family of proteins that play a key role in the regulation of various cell functions, including cell proliferation, differentiation, and survival [1]. These transmembrane receptors are integral components of cellular signaling pathways that modulate a wide range of physiological processes [1]. Dysregulation of RTKs, often caused by gain-of-function mutations, genomic amplifications, chromosomal rearrangements, or autocrine signaling mechanisms, has been implicated in the pathogenesis of numerous human cancers [2,3,4]. This abnormal activation disrupts normal cell regulation, leading to uncontrolled cell growth and disease progression [1,5]. Aberrant signaling of RTKs has been identified in more than fifty human RTKs in twenty subfamilies, contributing to the complexity and diversity of their pathological roles [1]. These dysfunctions are closely associated with the initiation and progression of various oncogenic and developmental disorders [6,7,8,9,10,11]. The dysregulation of RTK extends beyond oncology to neurodegeneration, psychiatric conditions such as depression and addiction, and genetic and developmental disorders [5].

Misregulation of RTKs through mechanisms such as overexpression, mutations, and cell-autonomous ligand–receptor interactions after loss of cell polarity underscores the critical need for targeted therapeutic interventions [5]. Pathological activation of RTKs is a key driver of tumorigenesis and metastasis, making RTKs a focal point for the development of targeted cancer therapies [12,13]. The emergence of small-molecule inhibitors and the approval of several of such agents for clinical use underscore the growing emphasis on RTK-targeted interventions in oncology [1,13]. RTKs are additionally involved in a variety of genetic and developmental disorders, including brain overgrowth syndromes that cause severe neurological impairments [5]. The disruption of the RTK signaling pathways is also associated with defects in the assembly or function of the primary cilia, resulting in a group of conditions known as ciliopathies [5]. Exploring RTKs in these contexts not only provides insight into their fundamental biological functions but also opens avenues for novel therapeutic approaches to address a wide range of diseases driven by RTK dysregulation [2].

The human FGFR2 protein (Uniprot P21802 · FGFR2_HUMAN), a member of the fibroblast growth factor receptor (FGFR) family, plays a crucial role in the RTK signaling network [14]. Structurally, FGFR2 shares the characteristic features of RTKs, including three immunoglobulin-like domains (Ig domains) in its extracellular region, a single-pass transmembrane domain, and an intracellular tyrosine kinase domain. Functionally, FGFR2 is involved in mediating cellular responses to fibroblast growth factors, influencing cell growth, survival, and differentiation. While mutations in human FGFR2, predominantly missense, occur across various domains of this protein, a significant portion of those with clinical implications are found in the FGFR2 tyrosine kinase domain, as this domain is central to the intracellular signaling cascade initiated upon binding of fibroblast growth factors to the extracellular domain of FGFR2. This dysregulation is associated with various cancers and developmental disorders, highlighting the importance of FGFR2 in the broader context of RTK-mediated diseases. The focus of this study is the FGFR2 tyrosine kinase domain. In the interest of convenience, we will refer to it from here on as FGFR2-TKD.

Mutations in FGFR2-TKD have been associated with craniosynostosis syndromes, such as Crouzon syndrome and Apert syndrome, highlighting the involvement of this receptor in skeletal disorders [15,16]. Mutations have been additionally implicated in promoting breast tumorigenicity by maintaining breast tumor-initiating cells [17] as well as in gastric, lung, endometrial, colorectal, and pancreatic cancers and in the development of drug resistance highlighting the importance of FGFR2-TKD in malignant neoplasms [14,17,18,19,20,21,22]. Elucidating the biological and clinical implications of aberrations/mutations in the broader FGFR family in paediatric and young adult cancers (and other disorders) is central for effective therapeutic treatment but currently challenging [23], in part because identifying patients most likely to benefit from FGFR inhibition currently rests on identifying activating FGFR mutations.

In FGFR2, the functional dynamics is key to clinical implications and consists of two primary states of FGFR2-TKD: an inactive, unphosphorylated state and an active, phosphorylated state crucial for signaling. Evidence of such states can be found in experimentally resolved tertiary structures of the FGFR2-TKD in wildtype and variant forms. Such structures, deposited in the Protein Data Bank (PDB) [24], show the receptor in its inactive and active structural states. These states are shown in Figure 1; the main structural alteration visually observed is that in the cytoplasmic region of FGFR2 (the activation loop colored in yellow).

Understanding the structural dynamics of FGFR2 is believed to be essential to the elucidating role of mutations in the prognosis of the disease, as well as in the development of targeted therapeutic interventions [25]. Hence, this paper formulates and seeks to address key questions on (i) how and to what extent do mutations in FGFR2-TKD impact the structural dynamics, (ii) how does the dynamics relate to functional dynamics, and (iii) how does mutation-impacted structural dynamics percolate to functional dynamics and then to potentially distinct disorders?

Contemporary predictive models of functional dynamics have traditionally focused on amino acid sequences alone, overlooking the critical role of the tertiary structure and therefore structural dynamics [26]. In this paper we take an integrative approach to connecting sequence variations to functional dynamics and disorders through structural dynamics. In particular, we utilize all known structures (deposited in the PDB) of FGFR2-TKD to capture and characterize the structural dynamics. Based on the principle of conformation selection [27,28], which in summary states that mutations alter the population probabilities of particular structures but do not remove from or add to the structure space, and a body of work in the Shehu laboratory capturing and characterizing the structure space of mutation-rich receptors [29,30,31,32,33,34,35,36], we seek to understand the biological and clinical implications of FGFR2-TKD mutations through a characterization of structural dynamics. In this paper we summarize the structure space documented in the PDB for FGFR2-TKD (over wildtype and variants) through machine learning methodologies that can capture both linear and nonlinear dynamics. The overlay of such dynamics over the sequence space allows us to identify regions that connect structure to function and provide insight on the functional implications of variant forms of FGFR2. In addition, connecting particular variants to diseases in which certain mutations are implicated allows us to identify disorder-specific signatures of dynamics, thus connecting structural and functional dynamics with disease.

**Figure 1 ijms-25-04523-f001:**
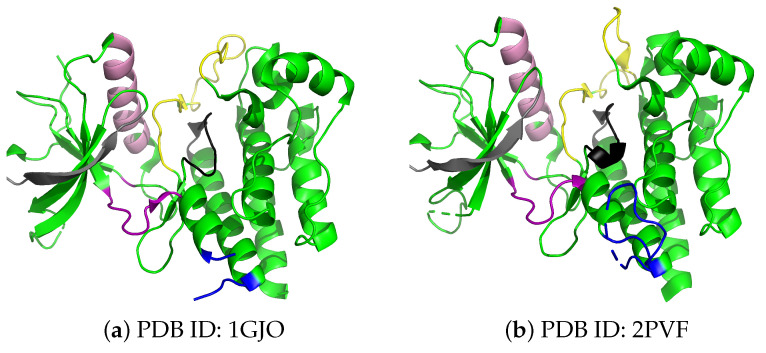
Structures of inactive and active forms of FGFR2-TKD. Panel (**a**) displays inactive FGFR2-TKD, highlighting the structural readiness of the kinase for activation, including the kinase activation loop and key unphosphorylated tyrosine residues. Color coding identifies the various known functional regions (see Table 1: nucleotide binding loop in gray, Alpha C helix in pink, kinase hinge in purple, kinase insert in blue, catalytic pocket in black, and activation loop in yellow). Panel (**b**) shows the active and phosphorylated FGFR2-TKD. The active structure is shown to accompany phosphorylation, enhancing substrate recognition and catalytic activity. The activation loop is distinct in the inactive and active structures, colored in yellow. The transition from an inactive to an active state is fundamental in various cellular processes, including growth and tissue repair.

**Table 1 ijms-25-04523-t001:** Distilling literature in [37,38,39], we identify the following functional regions: the nucleotide-binding loop, the Alpha C helix at the kinase N-lobe, the gate keeper, the kinase hinge (and its vicinity), the kinase insert, the catalytic pocket, the Alpha-C tether, and the activation-loop region. Among these, the activation-loop region is reported as the most critical for activation, especially the residues around position 659. In our collection of tertiary structure data from the PDB, we have removed chains missing any of these important regions. For more details, refer to Table 2.

Symbol	Region Name	Region Range
B	Nucleotide-binding loop	480–490
N	Alpha C helix at the kinase N-lobe	525–539
G	Gate keeper	564
H	Kinase hinge and its vicinity	566–571, 549, 565
K	Kinase insert	579–599
C	Catalytic pocket	620–630
T	Alpha-C tether	650
A	Activation loop	643–649, 651–664
O	Others	-

**Table 2 ijms-25-04523-t002:** Categorization of FGFR2-TKD Segment Structures by Activation State. The table provides a comprehensive categorization of 29 distinct tertiary structures of the cytoplasmic segment of FGFR2-TKD kinase sourced from the Protein Data Bank (PDB). These structures are classified into four primary activation states: active, inactive, intermediate, and unknown. The “unknown” category refers to structures whose activation status remains ambiguous because of interactions with inhibitory compounds, reflecting ongoing debates in the literature.

PDB ID (No. of Residue)	Status	Chains Removed
1GJO (316)	Inactive/WT	-
1OEC (316)	Inactive/WT	-
2PSQ (370)	Inactive/WT	-
2PVF (334)	Active/WT	-
3RI1 (313)	Inactive/WT	-
2PVY (324)	K659N/Active/A	$2PVY_{BC}$ removed, Missing residue 659
2PWL (324)	N549H/Active/K	-
2PY3 (324)	E565G/Active/K	-
2PZ5 (324)	N549T/Active/K	-
2PZP (324)	K526E/Active/N	-
2PZR (324)	K641R/Active/K	-
2Q0B (324)	E565A/Active/K	-
3B2T (311)	A628T/Active/C	-
3CLY (334)	C491A/Unknown/O	-
4J95 (324)	K659N/Active/A	$4J95_{AD}$ removed, Missing residue 654, 659
4J96 (324)	K659M/Active/A	-
4J97 (324)	K659E/Active/A	-
4J98 (324)	K659Q/Active/A	-
4J99 (324)	K659T/Active/A	$4J99_{A}$ removed, Missing residue 653
5EG3 (334)	Multiple/Unknown/O	-
5UGL (324)	D650V/Active/T	$5UGL_{B}$ removed, Missing residue 659
5UGX (324)	E565A/D650V/Active/HT	$5UGX_{B}$ removed, Missing residue 659
5UHN (324)	E565A/N549H/Active/H	$5UGN_{B}$ removed, Missing residue 659
5UI0 (324)	E565A/K659M/Active/HA	-
6LVK (313)	Unknown/WT	-
6LVL (313)	Unknown/WT	-
6V6Q (411)	Multiple/Intermediate/O	$6V6Q_{CD}$ removed, Missing residue 659
7KIA (308)	V564F/Unknown/G	-
7KIE (308)	V564F/Unknown/G	-
7OZY	2cMissing .PDB file	
8E1X	The reference paper information does not match with PDB structure.	

“Multiple” indicates that there are at least 8 mutation sites. For instance, PDB ID 5EG3 has 8 mutation sites, and 6V6Q has 10 mutation sites.

Another important contribution of this work is the enrichment of the analysis with computationally predicted tertiary structures; several FGFR2-TKD variants lack PDB-deposited tertiary structures. Utilizing and evaluating several methodologies, including AlphaFold2 and AlphaMissense, allows us to include in our analysis predicted tertiary structures. Connecting the structural and functional dynamics to disease then additionally allows us to elucidate potential functional and clinical implications of novel variants with no experimentally resolved structures.

The rest of this paper is organized as follows. In Section 4, we describe the methodological approach summarized above to connect FGFR2-TKD mutations to structural and functional dynamics and disease. Section 2 presents our findings. The paper concludes in Section 3 with a discussion of the major findings and future avenues of inquiry.

## 2. Results

We now embark on a comprehensive analysis, with a specific focus on the cytoplasmic region of FGFR2. This effort is rooted in the pioneering work of Chen et al., who provided valuable information on the molecular architecture of FGFR2 [37,38,39]. Our distillation of this literature reveals seven critical functional regions listed in Table 1, each playing a unique role in FGFR2 activity and regulation. These regions include the nucleotide binding loop (amino-acid residues 480–490), essential for ATP binding and energy transfer; the Alpha C helix (residues 525–539) in the kinase N lobe, which contributes to structural stability; the gate keeper residue at position 564, crucial for substrate specificity; the kinase hinge region (566–571) and its vicinity, which form a pivotal junction for domain movement; the kinase insert region, a segment known for its variability among kinases; the catalytic pocket (620–630), the site of enzymatic activity; the Alpha-C tether at position 650, a key element in maintaining the inactive state; and the activation loop (643–649, 651–664), which controls the transition between the active and inactive states. It is worth noting that the molecular brake is a key autoinhibitory mechanism predominantly located in the kinase hinge region of receptor tyrosine kinases (RTKs) [38]. It regulates kinase activity through a network of hydrogen bonds among three crucial amino acids, mediating autoinhibition when engaged and increasing kinase activity when disrupted by pathogenic mutations. This mechanism is directly linked with the development of various cancers and genetic disorders. Key functional regions affected by the molecular brake include: the kinase hinge region, where residues like E565, N549, and K641 form the core components of the brake; the Alpha C helix in the kinase’s N-lobe, which indirectly influences the molecular brake’s function, exemplified by the K526E mutation that creates new hydrogen bonds affecting the hinge area; and the activation loop, which contains multiple phosphorylation sites and whose conformation is directly impacted by the state of the molecular brake, indirectly regulating kinase activity through the stabilization of the A loop.

As detailed in Table 2, the PDB-deposited structures encompass a wide range of known states, thus offering a valuable dataset for our investigation. Our dataset not only incorporates these core structures but also includes an array of single-point mutations. These are key to understanding the alterations in the structural and functional status of FGFR2. The dataset is further enhanced by including structures with multiple point mutations, broadening our scope to explore a wide range of structural modifications. This is instrumental in examining how mutations affects the functional characteristics of the FGFR2 kinase.

In addition to linking structural dynamics to function, we also aim to obtain clinical implications of mutation-impacted dynamics in disease. We incorporate data from several databases, such as the Online Mendelian Inheritance in Man (OMIM), the Catalog of Somatic Mutations in Cancer (COSMIC), and the Human Gene Mutation Database (HGMD). Building on these databases, we have curated known disease categories and specific disorders under each category for FGFR2 variants with known and unknown structures, as listed in Table 3. Using these categories and diseases as labels, we analyze organizations of known (and predicted) tertiary structures of FGFR2-TKD wildtype and variants in a few dimensions that permit visualization of the structure space and so relate structural dynamics with disease.

### 2.1. From Structure to Function to Disease Linking Structure to Function to Disease for FGFR2 Variants with Known Structural Characterization

#### 2.1.1. Extracting Structural Dynamics

As described in Section 4, what we obtain from PCA are vectors/axes of a new space that that has been oriented to maximally capture the structural variation in a few dimensions. These vectors are the PCs. As also described in Section 4, the eigenvalue obtained with each PC from PCA measures the amount of data variation along that PC, and this variation can be employed to understand whether a few dimensions (a few PCs) are all that is needed to capture most of the variation in the data, that is, whether an intrinsic organization/dynamics emerges from the data in a few dimensions (as opposed to the close to 800 dimensions in the Cartesian space of CA traces of collected FGFR2-TKD tertiary structures).

Figure 2 relates the cumulative variance (measured as related in Section 4) as one includes PCs in the order of variation that they capture and shows the following: the first PC captures 67.09% of the structural variation, an indication that the structural variations are dominated by few concerted motions (in the PC space); with two PCs, one gets to 74.71% of the variation, and three PCs capture 81.00% of the variation. These results indicate that the top three PCs are meaningful and that focusing on these three alone misses less than 20% of the variation. Figure 2 also shows that only six PCs are needed to capture 90% of the structural variation; this level of efficiency in variance interpretation, particularly for an atomic distance matrix comprised of close to 800 dimensions (x,y,z coordinates for the CA atoms), underscores the robustness and practicality of PCA.

We can further evaluate the effectiveness of PCA in exposing the intrinsic structural dynamics of FGFR2-TKD by leveraging its high interpretability. As related in Section 4, we can “reconstruct” traces in the Cartesian coordinate space from projections on *i* PCs, where *i* can vary from 1 to higher values. Given the high variance captured by the top six PCs, we only focus on these in this exercise. In particular, we carry out the following exercise: Given a CA trace corresponding to a tertiary structure in our collected structures for FGFR2-TKD, we can obtain its projections over the top six PCs. So, each trace corresponds to a six-dimensional point in the space of the six PCs. If we zero out all but the first coordinate (the projection over PC1), each trace is now a one-dimensional point in the PC space. Per the equation in Section 4, inverting the projection and adding the average trace, we can obtain a corresponding (reconstructed) CA trace with each one-dimensional point. Measuring the RMSD between this reconstructed trace and the original one we can understand the reconstruction error resulting from ignoring the other five PCs. We can repeat this process now zeroing out all but the first two PCs, then all but the first three PCs, and so on. Figure 3 shows the reconstruction error (measured through the RMSD between the reconstructed and the original trace) as we include more PCs.

As expected, given that the top six PCs already capture more than 90% of the variance, the range for the RMSD-based reconstruction error is in the [0,1] Å. Also as expected, the RMSD decreases with an increasing number of PCs. Moreover, our findings reveal a significant decrease when considering the first three PCs, further suggesting that these three top components capture most of the structural variability in our dataset. It is worth noting that some tertiary structures (PDB ID 3CLY and PDB ID 6V6Q) show significant deviation (higher RMSDs) from this trend, representing outliers. The RMSD values for the structures with PDB IDs 3B2T and 6V6Q are elevated in the context of the first and second PCs when contrasted with other structures. This discrepancy can be attributed to the presence of additional structural elements or motifs within 3B2T and 6V6Q, potentially encompassing unknown functional regions or variations. These elements are not adequately captured by PCA and so are not reflected in the variation captured by the first two PCs.

The above quantitative results suggest four main findings: (i) there is an intrinsic structural dynamics of FGFR2-TKD, and it occurs in three major modes/dimensions; (ii) PCA is an effective methodology for exposing this dynamics; (iii) inspection of each of these three PCs and the CA atoms that they impact ought to now map this dynamics to the functional regions; and (iv) projection of the traces onto a few PCs ought to be informative and, when visualized, may provide useful information on potential co-localization of structures based on known functional and/or clinical implications.

#### 2.1.2. Linking Structural to Functional Variation

As related in Section 4, PCA is highly interpretable. For instance, we can visualize the magnitude of atomic displacements for each PC (we restrict here our attention to the top three PCs) over the amino-acid sequence, which allows us to understand how the dynamics is “distributed” over the functional regions.

As shown in Figure 4, the first three PCs greatly reflect the activation loop. This structural element is of paramount importance as it plays a decisive role in regulating the functional status of FGFR2. This loop plays an outsize role in the function of FGFR2, and our analysis captures this well. The structural variation present in the wildtype and variant forms of FGFR2-TKD is distilled well through PCA, and the top three modes of the intrinsic dynamics highlight the importance of this loop, connecting structure variation to function.

Furthermore, we observe that other functional regions, such as the kinase hinge and the nucleotide binding loop, although not predominantly featured in the first PC (PC1), are effectively captured in the second and third PCs. This pattern suggests that PC1 primarily encapsulates the most influential or critical functional regions of FGFR2. The absence of the kinase hinge and nucleotide binding loop in the primary component underscores the nuanced nature of protein dynamics, where linear methods like PCA may not fully capture the intricacies of non-linear structural changes. Moreover, this observation underlines the importance of examining multiple components to gain a comprehensive understanding of all functional dynamics within the protein. The selective prominence of these regions in PCs beyond the first also speaks to the need for broad analytical approaches that consider the complex interplay of structural elements across the protein. Another intriguing aspect of our findings is the prominence of residues in the 700 to 750 range within the PCA analysis despite these not being associated with a specific functional region. This could be attributed to the fact that these residues form a coil structure within the protein, which is inherently more unstable and thus exhibits more significant variations across different structures. Such fluctuations make this segment particularly distinct in our analysis, highlighting its unique role in the protein’s overall structure and behavior. This detailed analysis of intrinsic dynamics in FGFR2 not only corroborates the utility of PCA in protein structure analysis but also provides nuanced insights into the complex interplay of structural elements and their functional implications. The results not only validate established knowledge about critical regions such as the activation loop but also bring to light the dynamics of less prominent regions, offering a more holistic view of FGFR2-TKD structural dynamics.

The PC-based summarization of structural dynamics provides us with yet another approach to linking structural dynamics to functional variation; we can visualize the structural variation of wildtype and variant FGFR2 by projecting all structures onto the top two PCs. Color-coding each projection with the functional designation of their corresponding tertiary structure allows us then to potentially better infer functional implications of mutations. Figure 5 shows the 2D projection/embedding (over PC1 and PC2) of tertiary structures of wildtype and variant FGFR2-TKD. We observe that the PC1-PC2 projections effectively categorize mutations originating from the activation loop and kinase hinge regions, as we observe strong co-localization in the structure space. In addition to color-coded markers for different functional regions, we also introduce markers based on the activation status of the structure using different shapes. Circular markers represent inactive, triangular markers indicate active, rhombus markers signify uncertain status, and square markers represent an intermediate state, a status between active and inactive.

Incorporation of color and shape markers uncovers various clustering patterns. All active forms are well co-localized in various clusters in the 2D embedding. The PDB IDs 3CLY and 6V6Q remain distinctly separate from the others, aligned with our earlier observations of these structures as outliers. PDB ID 6V6Q is isolated, in agreement with the designation of this tertiary structure as in the intermediate state and unclear in functional implication. PDB ID 3CLY is also designated as unclear in both activity and functional region. 3B2T, the tertiary structure with a mutation in the catalytic pocket region, is distinct from the other variant forms and instead is co-localized with the wildtype inactive forms (PDB ID 1GJO and PDB ID 1OEC). This is due to the relatively conserved nature of the catalytic pocket region, located between beta sheets 6 and 8, and suggests that mutation of a catalytic pocket residue has minimal impact on the conformation of the activation loop. We also observe that 2PSQ, though labeled as ’inactive and unphosphorylated,’ lies away from 1GJO in the embedding. A detailed look reveals intriguing insights; PDB ID 2PSQ displays an activation loop region similar to the ’phosphorylated’ structure 2PVF. Despite its classification with inactive structures, 2PSQ’s spatial positioning in our analysis sets it apart from 1GJO and other unphosphorylated structures, hinting at a structural identity closer to the active state. This is consistent with previously published observations [37] where authors also notice that the structure is similar to inactive ones. It is worth noting that no published evidence points to 3RI1 behaving as 2PSQ. PDB ID 5UI0, notable for mutations E565A and K659M, which significantly increase kinase activity, resembles the activity level of wildtype phosphorylated FGFR2-TKD with an activated A loop. This dual mutation affects the allosteric network at two crucial points, resulting in enhanced kinase activity. However, as revealed by the PC1-PC2 embedding, 5UI0’s distinct placement sets it apart from other structures starting with 5’, such as 5UGX and 5UHN. Generally, structures prefixed with ‘4’ and ‘5’ are categorized as ‘active’, displaying various degrees of kinase activity. We also highlight the co-localization in the PC1-PC2 embedding of PDB IDs 7KIA and 7KIE, the only two tertiary structures with the gatekeeper mutation V564F which affects the ATP binding pocket and the activation loop, influencing the binding of kinase inhibitors and kinase activity, and with bound inhibitors.

It is worth highlighting that the two main clusters of the active forms of FGFR2-TKD positioned on opposite sides of the first principal component (PC1), as depicted in Figure 5, likely indicate substantial structural and functional diversity within these active states. This distribution suggests that PC1 captures the largest variance in the dataset, reflecting significant differences in activation mechanisms, such as variations in activation loop conformations, domain orientations, or Alpha C helix positioning. Additionally, these clusters may highlight functional diversities within active states, potentially influenced by ligand binding, post-translational modifications, or specific mutations that confer activation, each leading to unique structural alterations. The distinct clusters could also be a result of data-driven separation due to the conformational sampling present in the structural datasets used for PCA, with the protein adopting multiple active conformations throughout its functional cycle. The separation into clusters thus reflects the complex interplay of structural adaptations, functional requirements, and external modulatory factors that characterize the dynamic nature of protein activation and regulation.

As related in Section 4, though not as interpretable as PCA, we also employ Isomap to extract the intrinsic dynamics of FGFR2-TKD. While Isomap does not reveal the amount of variation over amino acids per its components, projections of tertiary structures over the top two components can nonetheless be obtained and labeled as above, shown in Figure 6. Figure 6 reveals that Isomap is closely aligned with PCA in classifying the major functional regions. Both techniques successfully distinguish between active activation loop structures and wildtype inactive structures and isolate uniquely dissimilar structures, such as 3CLY and 6V6Q. Compared to PCA, Isomap better separates mutations in the gatekeeper region (in red) and the kinase hinge region (in purple). However, Isomap is not as effective as PCA in segregating activation loop mutations; Isomap places 2PSQ closer to 1GJO despite 2PSQ’s activation region being more akin to a phosphorylated state.

#### 2.1.3. Linking Structural Variation to Disease Implication

Table 4 recapitulates the functional status of each structure and adds the disease classification where relevant. It is worth noting that not all the mutations/combinations of mutations were observed clinically; that is, the mutations that were “ported” from FGFR3.

The 2D embeddings of tertiary structures, whether obtained with PCA or Isomap, can now be re-analyzed by labeling projections of structures with their disease classification. Figure 7 shows this analysis for the PC1-PC2 embedding. A clear segregation is observed between most of the genetic disorders and cancers, underscoring the role of structural dynamics in disease differentiation. Notable exceptions are “unclassified Craniosynostosis Syndrome” and “LADD1 syndrome”, represented by red and pink, respectively. Interestingly, both the UCS structures have the same mutation (K659N) that was identified in a patient not initially recognized as having Cruzon-like features [40]. 3B2T was previously recognized as similar to wildtype FGFR2 [41], LADD syndrome is clinically distinct from Pfeiffer and Crouzon syndromes, and the A628T mutation, in the catalytic pocket, decreases kinase activity rather than activates it. It is worth noting that the ambiguity re UCS and LADD1 can also be attributed to the limited sample size available for these conditions, highlighting the challenges with understanding the molecular basis of rare disorders.

Figure 8 relates the analysis over the Isomap-obtained embedding. The Isomap embedding shown in Figure 8 does not exhibit superior classification when compared to the PCA projection and does not separate well between developmental and oncogenic disorders.

The lack of experimentally available structures motivates us to enrich the analysis with computationally predicted ones, as we relate next.

### 2.2. Extending the Analysis with Computational Models of Variants with No Structural Characterization

As related in Section 4, we pursue three distinct structure prediction methods. First, we evaluate the faithfulness of these methods in reproducing known tertiary structures of FGFR2-TKD; we limit this analysis to single-nucleotide variants, of which there are 13 in the list of 29 with known tertiary structures listed in Table 2. For each, we measure the RMSD between the known and the predicted structure for each of the three methods (SWISS-MODEL, AlphaFold2, AlphaMissense) and relate it in Figure 9. All three methods are effective at reproducing known structures, as their error is below 0.8 Å for each structure. However, the method with the lowest RMSD per structure is SWISS-MODEL, followed by AlphaMissense and then AlphaFold2.

The difference between a 0.4 Å RMSD (which is what SWISS-MODEL achieves on each structure) and a 0.4–0.8 Å RMSD (which is the range for AlphaFold2) is made evident in Figure 10, which zooms in on the predicted and known structure for PDB ID 2PZ5.

Figure 10 offers several insights. AlphaFold2, despite its sophisticated algorithmic approach, faces challenges in coil regions. This issue could be attributed to potential biases within its training dataset, which might favor certain structures over others. In comparison, homology modeling through SWISS-MODEL displays a higher degree of adaptability to various structural states. This method seems to offer a more nuanced understanding of the protein structures, particularly in capturing the range of structural variations. It is also worth noting that both AlphaFold2 and AlphaMissense, while offering groundbreaking perspectives in protein modeling, show a noticeable decline in predictive accuracy in less structured regions like the activation loop in coil areas. Our findings underscore the need for a careful and critical approach when interpreting the results in these specific regions. The distinct methodologies, although robust in many aspects, require cautious application and interpretation, especially in the context of structurally complex and dynamically variable regions such as the activation loops within protein coils.

Taken altogether, the above analysis demonstrates the superiority of SWISS-MODEL in our case. The SWISS-MODEL-generated structures closely match the original PDB structures. The structures, further filled (with SCWRL4) and refined with NAMD2, exhibit a high degree of similarity to the actual structures in the PDB. Based on these observations, we decided to employ SWISS-MODEL homology modeling for all subsequent mutant modeling tasks.

Having established SWISS-MODEL to be a more faithful structure prediction method in our context, we apply it to variant sequences with no known structures. Followed by SCWRL4 and the energetic refinement described in Section 4, we now enrich our prior embedding-based analysis with computed (and refined) structures.

Figure 11 shows the PC1-PC2 embedding of known and computed tertiary structures with colors and shapes utilized to indicate functional region and state designations. ’X’ is utilized to further indicate the computed structures. Figure 11 suggests that R6127 and R678G are distinct, though possibly both in an active state, whereas A648T may represent an intermediate, unclear state in terms of activity.

Figure 12 now adds the computed structures to the PC1-PC2 embedding labeled with disease classifications. Some interesting observations emerge. Modeled structures with the mutations R678G (associated with Crouzon syndrome), A648T (associated with LADD1), G663E (associated with Pfeiffer syndrome), and R612T (associated with lung cancer) have been meticulously classified into specific disease-related categories. In particular, R678G and G663E align closely with mutations causing genetic/developmental disorders, showing a strong correlation with this group. On the contrary, A648T and R612T show a less precise alignment with their respective groups; A648T is somewhat close to the 3B2T marker, suggesting a less definitive association, while R612T aligns more loosely with cancer-causing mutations. It should also be noted that R612T is distinct in its position in the embedding, which is consistent with being the only mutant with a classification of lung cancer.

We note that the A628T (PDB entry 3B2T) mutation affects the intrinsic catalytic activity of FGFR2 kinase by compromising the catalytic activity of the kinase domain. The A628T mutation alters the configuration of key residues in the catalytic pocket, leading to decreased tyrosine phosphorylation of FGFR2 and recruitment of downstream signaling molecules. Similarly, A648T confers a loss of function demonstrated by decreased proliferation relative to wildtype FGFR2 in a competition assay and decreased transformation activity [42], decreased protein kinase activity, and reduced downstream MAPK signaling pathway activation in cultured cells. Both mutations are reported with LADD syndrome, even though the mutation positions are not located in the same functional region.

## 3. Discussion

Elucidating the biological and clinical implications of mutations in proteins central to human health is challenging but key to effective, targeted, personalized therapeutics. In this paper, we have focused on FGFR2, a key protein in regulating cell proliferation, growth, and differentiation. Decades of computational and experimental research on the relationship between sequence, structure, and function have demonstrated the importance of structural dynamics in this relationship [26]. In particular, a key realization is that mutations often impact the structural dynamics of a protein and through it they impact the ability of the protein to carry out its “wildtype” interactions with other molecules in the cell, percolating into disrupted chemical pathways that then demonstrate clinically in disorders and disease [43].

We leverage this realization in this paper and aim to characterize and summarize the structural dynamics of FGFR2. By relying on the principle of conformation selection [27,28], we harness the structural diversity present among PDB-deposited tertiary structures of wildtype and variant forms of FGFR2 in various functional states. Through various machine learning methodologies we reveal intrinsic organizations of the structure space populated by the experimentally resolved structures that expose a few intrinsic modes of dynamics. The methodological approaches presented here provide a mesoscale global view of the structural states that highlight major modes of motion and parts of the structure with coordinated movements. It complements the atomic resolution results from molecular dynamics and X-ray crystallization studies that revealed changes in, for example, H bonding patterns of the molecular brake residues between the active and inactive states.

Through various structure prediction methods, we enrich this analysis with computationally modeled tertiary structures of FGFR2 variants currently with no structural characterization. Mapping the identified intrinsic modes of structural dynamics on the amino-acid sequence of FGFR2 and expanding this to wildtype and variant sequences reveals which functional regions are most implicated by the dynamics and in which variants and functional states, drawing a connection between sequence mutations, structural dynamics, and functional dynamics. Labeling the elucidated organization of the structure space with disorder information further completes the sequence–structure–function relationship by bringing into it clinical implications.

What have we learned about FGFR2 from this analysis? Several observations emerge. The dynamics captures well the prominent, outsize role of the activation loop in all the top intrinsic modes of dynamics. Other functional regions, such as the kinase hinge and nucleotide binding loop also feature prominently in the top modes. In particular, the detailed analysis of the intrinsic dynamics in FGFR2 in this paper not only corroborates the utility of PCA but also provides nuanced insights into the complex interplay of structural elements and their functional implications. For instance, the results presented earlier not only validate established knowledge about critical regions such as the activation loop but also expose the dynamics of less prominent regions, thus offering a more holistic view of FGFR2-TKD structural dynamics. In addition, the identified dynamics provides good co-localization of active forms of FGFR2 and even provides us with a better understanding of subtle structural indications in a tertiary structure documented as inactive in the PDB but with structural indications that correctly place it closer to active structures in our analysis.

The presented approach is able to distinguish the partially inactivated structure of 2PSQ from other structures in the autoinhibited state, a finding that had been overlooked by many previous studies. Future studies need to confirm our finding of 3RI1 behaving as 2PSQ. The results suggest that disease-associated mutations differ not only in their location in the protein sequence but also in their impact on the 3D structure. In addition to previously recognized differences in the activation loop for activating mutations, the models distinguished activating mutations in different regions and the inhibitory mutations associated with LADD syndrome, although they lie in different subregions of the protein. RTKs are a large family with many members of clinical importance and that share dysregulation of kinase activity as a disease mechanism. Future work will explore whether the approach applied here to FGFR2 can also contribute to the characterization of structure–function–disease relationships for other RTKs.

There are several limitations to our approach. First, it relies on crystal structures determined by different laboratories, using different methods, and in different space groups. Batch effects such as these could affect the 3D structures. We are unable to distinguish whether series of structures are more similar to each other because of their functional relationship or because of unaccounted for effects. Second, while the results suggest that there is some correlation between the functional impact of mutations and the associated disease, protein structural methods cannot account for other factors that contribute to the manifestation of the disorder, including clinical factors such as whether the mutation is germline or somatic. Indeed, some mutations could be associated with either germline or somatic diseases.

The results from using molecular models for mutations without existing protein structures shows promise for application to novel mutations with unclear mechanism of action. Previous studies have suggested that activating mutations function by changing the equilibrium between activated and inactivated forms. The method applied here captures snapshots of the structure space rather than the relative frequency of each structure but suggests that in addition to the balance of activated and inactivated forms of the protein, there may also be structural differences between different activated states or their intermediates.

Interesting additional observations emerge with regards to the impact of mutations on disease. We note that in addition to the analysis we have presented in the main paper, we relate here a complementary analysis on the ability of AlphaMissense to capture the pathogenicity or not of sequence mutations. While we have evaluated AlphaMisense in the main paper in its ability to reproduce known tertiary structures, AlphaMissense is a larger computational system that also predicts pathogenicities (or not) given a protein sequence. In Table 5 we relate the scores and predictions from AlphaMissense on the potential pathogenicities of all FGFR2 variants in our dataset (the wildtype is correctly predicted as benign). Column 4 discretizes these scores into hits or not and shows that AlphaMissense correctly predicts that the FGFR2 variants we study are pathogenic in 14/16 cases (87.5% of the time).

We are not yet at the point where we can predict the actual pathogenicity from a given protein amino-acid sequence, but we are making progress, as the findings in this paper relate. We note among our findings that the identified structural dynamics for FGFR2-TKD separates well between developmental disorders and oncogenic disorders and reproduces the clinical implications of FGFR2 variants with no known structural characterization (for which we employ computationally modeled structures) from their positions in the PCA-reduced structure space. Further fine-grained differentiation suffers due to an imbalanced distribution of tertiary structures among disorders.

The availability of more tertiary structures in various forms and clinical settings for FGFR2 will further help the discriminative power of the methodology presented in this paper. As related in the detailed analysis, already we see its promise in providing a complete picture that brings together sequence, structure, dynamics, function, and clinical implications for a holistic understanding of the molecular basis of a disorder and so targeted therapeutic treatments.

### Methodological Considerations

Understanding the structural basis of protein function and its alteration due to pathogenic mutations is crucial for the development of targeted therapeutic strategies. In our study, we primarily focused on Principal Component Analysis (PCA) and Isomap due to their broad applicability and robustness in capturing structural dynamics. However, we recognize the importance of discussing the strengths and limitations of these and other relevant methods in the context of pathogenicity analysis.

PCA is advantageous for its simplicity, interpretability, and efficiency in identifying the principal axes of variation within a dataset. Its linear nature allows for the straightforward projection of new data onto the derived principal components, facilitating the comparison of structural variants. However, PCA’s linear approach may not capture complex, non-linear relationships within the protein structures, potentially overlooking subtle but critical variations associated with pathogenicity.

Isomap, on the other hand, excels in mapping non-linear manifold structures, making it particularly useful for uncovering intricate structural dynamics that linear methods like PCA might miss. This capability enables a deeper understanding of the protein’s conformational space and its alteration by mutations. The main drawback of Isomap is its computational complexity and the difficulty of interpreting the resulting dimensions, which can limit its utility in large-scale or rapid analyses.

Other methods, such as t-Distributed Stochastic Neighbor Embedding (t-SNE) and Uniform Manifold Approximation and Projection (UMAP), have also been employed to explore structural diversity and pathogenicity. t-SNE is highly effective in visualizing clusters of high-dimensional data but is computationally intensive and sensitive to parameter settings, which can affect reproducibility. UMAP offers a balance between t-SNE’s detailed visualization capabilities and PCA’s scalability, providing high-quality embeddings with less computational demand. However, both t-SNE and UMAP can produce embeddings where the relative distances between points do not always represent meaningful biological relationships, potentially complicating the interpretation of pathogenic versus non-pathogenic structural variations.

In summary, each method has its unique strengths and limitations in the context of studying protein structural diversity and pathogenicity. Our choice of PCA and Isomap was guided by the objectives of our analysis, the nature of our dataset, and the need for a balance between computational efficiency and the ability to capture complex structural dynamics. We acknowledge the importance of considering a range of computational tools for a comprehensive understanding of the structural bases of protein pathogenicity and encourage future studies to explore the complementarity of these methods in greater detail.

## 4. Methods and Materials

### 4.1. Data Collection and Preparation

As related earlier, we leverage experimentally resolved tertiary structures in the PDB. Searching for the cytoplasmic region of FGFR2-TKD returns 29 distinct tertiary X-ray structures, as listed in Table 2. The PDB IDs of these structures and their corresponding residue length are listed in Column 1. The functional status of each structure, together with information on whether the structure belongs to a wildtype or a variant, in which case the missense mutation is also listed, is provided in Column 2. Such information is obtained from the authors’ classification of the structure as representing activated or inactive forms in the publication that accompanies the PDB entry submission. Column 3 provides details on the structural integrity and the chains that we removed prior to our further analysis for lack of structural integrity (e.g., missing residues). After removing such chains, we end up with 52 chains collected from 29 tertiary structures.

Prior to characterizing the structural dynamics, the dataset needs to be processed. First, the input structures can have a different number of atoms. This is the case due to the dataset containing different variants of a protein molecule as well as due to intrinsic differences on the level of structural detail provided in different PDB entries; for instance, the number of side-chain atoms for which Cartesian coordinates are provided can vary greatly among PDB entries of the same protein molecule. To resolve this first source of variation, we focus only on the main-chain carbon (CA) atoms. That is, from each PDB entry, we extract only the CA atoms, and now each tertiary structure in our dataset is a vector of Cartesian coordinates for the CA atoms. These are often referred to as CA traces. Given 263 CAs, each trace is then represented as a vector of 263 × 3 = 789 Euclidean (x, y, z) coordinates.

Second, the input structures can be in different poses in 3D due to rigid-body motions; that is, translation and rotation in three dimensions (3D). This can be addressed by aligning all tertiary structures (vectors of CA coordinates/CA traces) onto a reference CA trace. The CA traces are aligned to some reference trace using the optimal superimposition process typically employed when identifying least root-mean-square deviation (lRMSD) between two structures [44].

We deliberate on the choice of the reference trace, though often in computational work that is chosen arbitrarily. In particular, we select the CA trace from the tertiary structure under PDB ID 1GJO as our reference trace. In evaluating the suitability of PDB IDs 1GJO, 1OEC, 2PSQ, and 3RI1 as likely candidates for a representative for an unphosphorylated FGFR2-TKD (and so a reference trace), it becomes evident that PDB ID 1GJO emerges as the most appropriate choice. The crystal structure of unphosphorylated WT FGFR2-TKDK (2PSQ) demonstrates a structure inconsistent with a genuine inhibited state. This is evidenced by the similarity of the location of the Alpha C helix and the A-loop conformation in 2PSQ to those in phosphorylated and mutationally activated FGFR structures. Furthermore, the accessibility of the substrate tyrosine binding site in 2PSQ contrasts starkly with genuinely inhibited structures where an A loop plug occludes the active site. This implies that 2PSQ does not adequately represent the inhibited conformation typical of unphosphorylated FGFR2-TKD [37]. Meanwhile, PDB ID 3RI1, which has two ligand bindings, deviates from the conventional unphosphorylated structure, rendering it unsuitable for this purpose. Regarding 1OEC and 1GJO, both share similarities, but 1GJO is preferred due to its superior resolution. Higher resolution in crystal structures is indicative of more accurate atomic position data, making 1GJO a more reliable reference for the unphosphorylated state of FGFR2-TKD and so an ideal reference trace.

### 4.2. Characterization of Structural Dynamics Present in Experimentally Resolved Tertiary Structures

Once the traces are aligned onto the reference trace, we then subject them to two different machine learning methodologies that effectively take the structure space (in Cartesian coordinates) represented by the traces and map it into a different coordinate space that often better exposes the organization of structures and summarizes the source of structural variation (hence, the structural dynamics) through a few coordinates. Since such methodologies can take into account linear or nonlinear dynamics, we employ two distinct representatives, Principal Component Analysis (PCA) [45] versus Isomap [46]. Below, we summarize these methodologies and the way we employ them to reveal the intrinsic dynamics.

#### 4.2.1. PCA for Characterization of Linear Structural Dynamics

We employ the PCA implementation in the Python (3.11.4) module sklearn.decomposition. PCA [47], a choice motivated by its robustness and ease of integration into our analytical pipeline. The aligned traces are deposited in a matrix of 52 rows corresponding to the number of CA traces related above and 789 columns corresponding to the x,y,z coordinates of each trace/row. An average trace (AverageTrace) is computed and subtracted from all the traces, resulting in a centered matrix *X*. The *X* matrix is then subjected to a singular value decomposition $X=U\cdot \Sigma \cdot V^{T}$. The new axes or principal component (PCs) are the rows of the *U* matrix, and the singular values, which are the square roots of the eigenvalues corresponding to the PCs, are the diagonal entries of the Σmatrix. The PCs are ordered from largest to smallest corresponding eigenvalue; an eigenvalue measures the variance captured by the corresponding PC if the data (traces aligned and centered) are projected onto it. This process reflects the fact that PCA transforms the input data in such a way that maximizes the data variation in few dimensions/PCs. PCA is a linear dimensionality reduction method. That is, it cannot capture any present nonlinear dynamics. But it provides great flexibility and interpretability. A new structure *S* (that is, its CA trace and post-alignment onto the reference trace), even if not included in the PCA, can be readily projected onto the space of extracted PCs. Its projection can be obtained using the equation $P_{S}=\left(S-\mathrm{AverageTrace}\right)\cdot U$. Even more interestingly, an aligned CA trace *S* can be recovered from a projection $P_{S}$ through the equation $S=P\cdot U^{T}+\mathrm{AverageTrace}$. These two equations are important both to subject new structures (for instance, computationally modeled structures), as well as visualize structural changes along specific dimensions by generating traces where only one PC is varied versus others to understand what portions of the structure a PC varies and so controls. This allows us to connect PC-guided structural dynamic to functional regions.

In the realm of structural biology, the application of PCA to understand protein dynamics is well established. The seminal work by Amadei and colleagues introduced the concept of essential dynamics, demonstrating that the majority of protein dynamics can be captured by a few principal modes of motion, a principle that underpins our approach [48]. Similarly, Maiorov, and Crippen emphasized the significance of root-mean-square deviation (RMSD) in comparing protein structures, which is crucial for evaluating the accuracy of our PCA-based structural reconstructions and ensuring that the identified principal components genuinely represent the protein’s structural dynamics [49]. Further supporting our use of PCA, Kitao and Go explored protein dynamics in collective coordinate space, highlighting PCA’s utility in identifying the essential dynamics critical for protein function and interaction [50].

The power of PCA (and other dimensionality reduction methods) is that the transformation they provide (from the original space onto the new space of PCs *U*) allows exposing how much structural variance each new axis vector (PC) captures. This information is provided by the corresponding eigenvalue of each PC. Normalizing these eigenvalues allows us obtaining structural variation as a percentage of the total variation (100%) captured by all the PCs. So, a cumulative ranking where one can visualize the number of total variation captured by adding PCs (PCs are considered in the order of highest to lowest corresponding eigenvalue) exposes the number of PCs needed to capture a target structural variation. For instance, such analysis, as related in Section 2, can reveal that few PCs are needed to capture 80% structural variation (the structural dynamics. This is the power of dimensionality reduction methods such as PCA. The other benefit is, as related above, that we can visualize directly which aspects of structural dynamics each of the PCs capture and so can map the structural dynamics to functional dynamics. Moreover, visualizing projections of the traces onto a few PCs and color-coding projections based on functional or disease characteristics of the corresponding tertiary structures allows further understanding the relationship between structural dynamics, functional dynamics, and disease. These analyses are related in Section 2.

#### 4.2.2. Isomap for Characterization of Nonlinear Dynamics

Isomap, which stands for Isometric Mapping, can capture any intrinsic nonlinear dynamics. Unlike PCA, which aims to preserve the variation of the data, Isomap aims to preserve the pairwise geodesic distances between data instances. First, a nearest-neighbor graph is constructed from the data. Given a user parameter *k*, the *k*-nearest neighbors of a data instance are determined using Euclidean distances. Once such calculations are completed for all data instance, the nearest neighbor graph then connects with an edge every data instance to its *k* nearest neighbors. Unlike a centered matrix *X* as in PCA, Isomap constructs a matrix of geodesic distances between data instances utilizing the nearest-neighbor graph. That is, given n=52 data instances, an n×n matrix is constructed. An entry at row *i* and column *j* stores the shortest path computed over the nearest-neighbor graph connecting data instance *i* to data instance *j*. This matrix is subjected to Multidimensional Scaling (MDS) to find a low-dimensional embedding that best preserves these pairwise distances. MDS is a distance-preserving dimensionality reduction method; using techniques like eigenvalue decomposition or optimization algorithms, MDS reduces the discrepancy between the original pairwise distances and the distances in the low-dimensional space. We employ the Isomap implementation in the Python module sklearn.decomposition.PCA [47]. We use the recommended/default value for k=8. Like PCA, Isomap can be utilized to visualize the original structure space (the CA traces) onto the few axes revealed by the MDS. However, while in principle able to better capture relations among data, Isomap lacks flexibility and interpretability. A new trace cannot be readily projected onto the obtained axes. It needs to be included in the nearest-neighbor graph, and the computations summarized above have to be repeated. Isomap is also not interpretable. There is no ready mechanism through which we can understand what aspects of dynamics each of the revealed new dimensions/axes captures, thus limiting our ability to map structural dynamics to specific functional regions. It is worth noting that for all plots where we aim to link structural dynamics (whether linear through PCA or nonlinear through Isomap) to functional regions, activity, or disease, we utilize the matplotlib [51] library in Python, a versatile tool for creating high-quality figures and plots in a Python environment.

### 4.3. Computational Modeling to Predict Novel Tertiary Structures In Silico

We observe that for several mutations of FGFR2-TKD, there are no experimentally resolved tertiary structures. We utilize three different structure prediction methods (a homology-based method, AlphaFold2, and AlphaMissense) to model these missing structures. More importantly, we carry out an evaluation of these methods onto known structures to see to what extent they can reproduce existing, known structures to determine a method that is more faithful and so more reliable to be used in modeling unknown structures. Once we determine the best method, we energetically fine-tune the predicted structures and then include them in our structural dynamics characterization summarized above. This allows us to derive functional and disease information for several mutations, currently with no wet-lab structural characterization.

#### 4.3.1. Structure Prediction

##### SWISS-MODEL

We utilize SWISS-MODEL (https://swissmodel.expasy.org/, accessed on 12 January 2024) [52] as a representative of homology-based structure prediction methods. SWISS-MODEL is a web-based tool developed by the Biozentrum University of Basel and the Swiss Institute of Bioinformatics that infers the structure of a given amino-acid sequence based on its alignment with sequences with known tertiary structures. For our analysis, we submitted the primary amino acid sequence (of a target) to SWISS-MODEL, which then performed a comparative analysis against its extensive database of known protein structures. The platform algorithm meticulously selected the most suitable structural templates and constructed a model by aligning the target sequence with these templates, adhering to the default settings for sequence-based modeling. These settings are optimized to balance accuracy and computational efficiency, enabling the generation of high-quality models without the need for manual adjustment of the parameters. Our selection of SWISS-MODEL was motivated by its demonstrated precision, reliability, and comprehensive integration of the latest advances in computational structural biology.

##### AlphaFold2

In our study, we incorporate AlphaFold2 (https://colab.research.google.com/github/sokrypton/ColabFold/blob/main/AlphaFold2.ipynb, accessed on 23 November 2023) [53], a revolutionary protein structure prediction tool developed by DeepMind Technologies. AlphaFold2 represents a significant advance in the field of computational biology, using an innovative deep learning approach to predict tertiary protein structures with remarkable precision in the absence of highly similar amino-acid sequences (that is, beyond the realm of homology modeling). Given an amino-acid sequence and using default parameter settings, we obtain from AlphaFold2 6 structures from which we utilize the top-ranked structure for further analysis. The rank reflects the amount of confidence with the prediction.

##### AlphaMissense

We additionally consider AlphaMissense [54], a recent method proposed to assess the impact of missense mutations on protein structure. Key to AlphaMissense is the prediction of tertiary structure of a single amino-acid chain. Rather than just utilizing AlphaFold2 for this component, AlphaMissense trains a network similar to that in AlphaFold2 along with protein language modeling by predicting the identity of the amino acids masked at random positions in the MSA. The result is a slightly different methodology for tertiary structure prediction, as well, which we leverage here and refer to as AlphaMissense in our evaluation analysis in Section 2.

##### Structure Completion and Energetic Refinement

In Section 2 we evaluate these three methodologies for their ability to reproduce known tertiary structures. By effectively comparing the RMSD between predicted and known structures for each method, we select one most reliable method which we then utilize to predict novel tertiary structures. To improve the energetic profile of the predicted structures, we first complete the predicted structures to contain atomic coordinates for all side-chain atoms, information typically not obtained from structure prediction methods. We utilize the popular SCWRL 4.0 [55] from the Dunbrack lab for this purpose. This software tool utilizes a backbone-dependent rotamer library, specifically, the Dunbrack rotamer library, and leverages CHARMM as the scoring function to evaluate side-chain conformations, focusing on minimizing steric clashes and optimizing hydrogen-bonding interactions. The resulting all-atom structure obtained is then subjected to an energy minimization protocol. We employ NAMD2 [56], which utilizes CHARMM as the force field, applying fixed boundary conditions, and employing Particle Mesh Ewald (PME) for the treatment of long-range electrostatic interactions. The minimization protocol is set to a length of 10,000 steps. This optimization process is vital for mapping a structure to its nearby lowest-energy state, ensuring its physical realism and stability in any potential dynamic simulations thereafter. Section 2 employs these predicted and energetically refined structures to derive functional and disease information for several variants. We now relate the findings of our study.

## Figures and Tables

**Figure 2 ijms-25-04523-f002:**
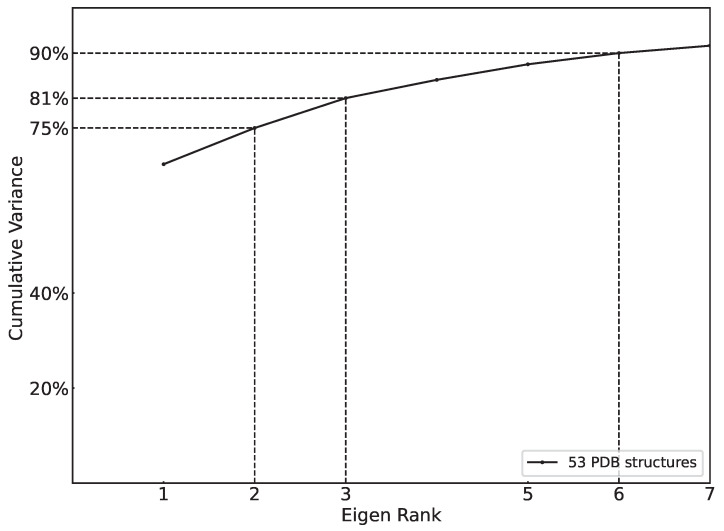
The cumulative variance analysis shows that PCA is effective for FGFR2 structural variation. The plot stops after the top 7 PCs, as 6 PCs already capture more than 90% of the variance.

**Figure 3 ijms-25-04523-f003:**
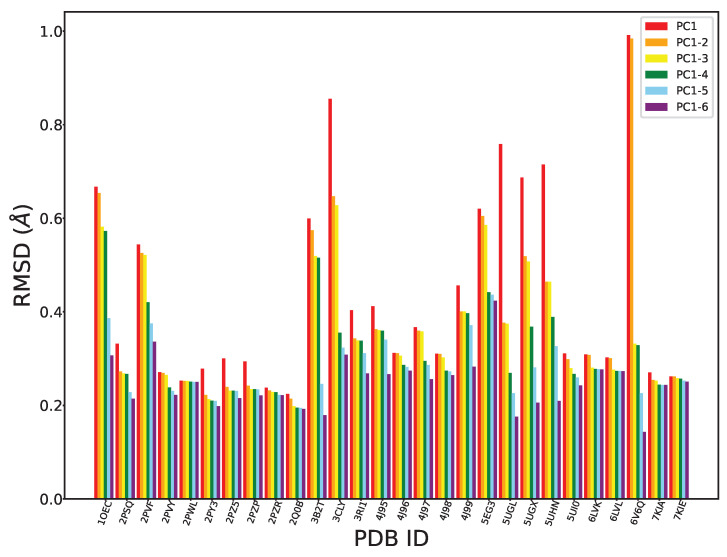
This plot shows the RMSD between the original CA trace (subjected to the PCA analysis) and the corresponding reconstructed CA trace as a function of including more PCS. RMSD values range from zero to one in the plot, which accounts for the relatively smaller variations observed. We observe that 3CLY and 6V6Q exhibit distinct characteristics compared to the other structures.

**Figure 4 ijms-25-04523-f004:**
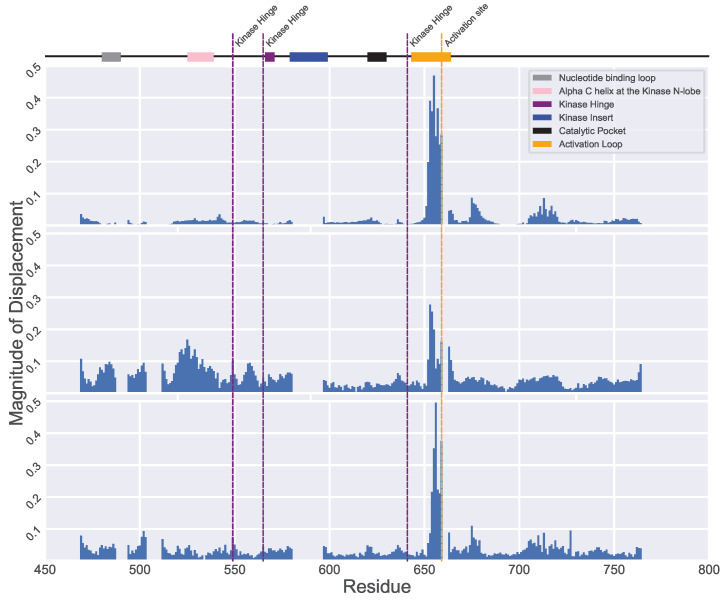
We map out the magnitude of dynamics over the amino acids for each of the top three PCs. The top panel shows displacements along PC1; the middle panel does so along PC2, and the bottom panel shows displacements along PC3. Each of the PCs effectively highlight the activation loop, a pivotal structural element influencing FGFR2’s status and function, as expected. Furthermore, while PC1 shows less prominence for regions like the kinase hinge and nucleotide binding loop, PC2 and PC3 capture them well. Residues between 700 and 750, lacking a specific functional role, are prominently featured due to their inherently unstable coil-like structure, leading to distinctive variations across structures. The catalytic pocket is not adequately captured. This limitation comes from the fact that we have only one structure, PDB ID 3B2T, with a mutation in this region, making it challenging to draw robust conclusions about its behavior.

**Figure 5 ijms-25-04523-f005:**
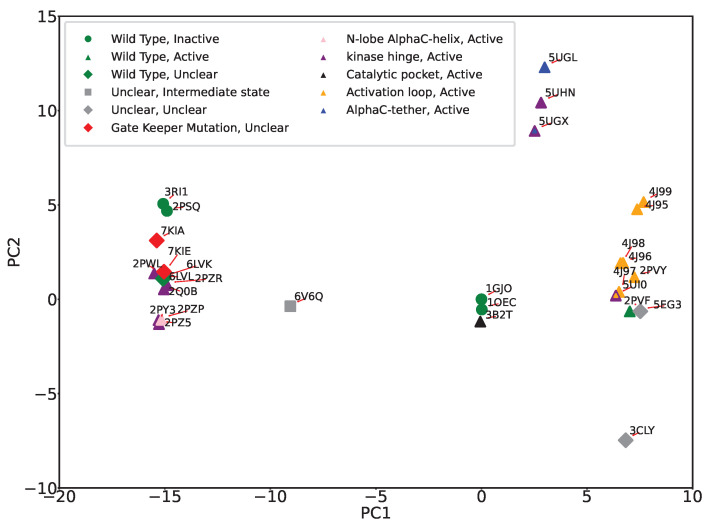
The 2D (PC1-PC2) embedding of tertiary structures of wildtype and variant FGFR2-TKD is labeled based on functional regions (colors) and activation status (shapes). All active forms are well co-localized in the 2D embedding. Red arrowed lines are added to improve visibility by better associating PDB IDs with the color-coded markers. Several interesting clustering patterns emerge, as detailed in the main text.

**Figure 6 ijms-25-04523-f006:**
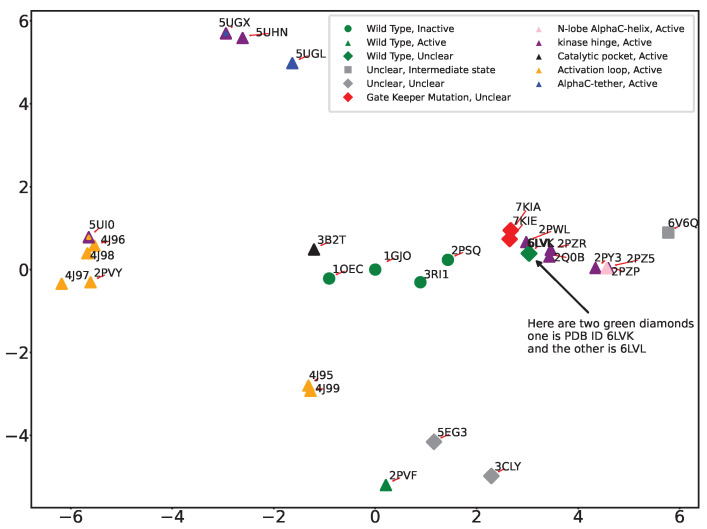
The Isomap-obtained 2D embedding reveals notable shifts in the positioning of PDB IDs 3RI1 and 2PSQ relative to the reference structure 1GJO. Specifically, 2PSQ, despite its activation loop resembling a phosphorylated state, largely mirrors an unphosphorylated structure, leading to its placement near the inactive wildtype cluster. Conversely, 3RI1, characterized as an unphosphorylated FGFR2 kinase domain, is displayed in an auto-inhibited state. This finding is significant for identifying potential targets for selective inhibitors. Isomap effectively isolates distinct structures such as 3CLY and 6V6Q from the rest. Moreover, it categorizes structures with the prefixes “4”, “5”, and “2”, elucidating their interrelationships and contributing to our understanding of structural variations.

**Figure 7 ijms-25-04523-f007:**
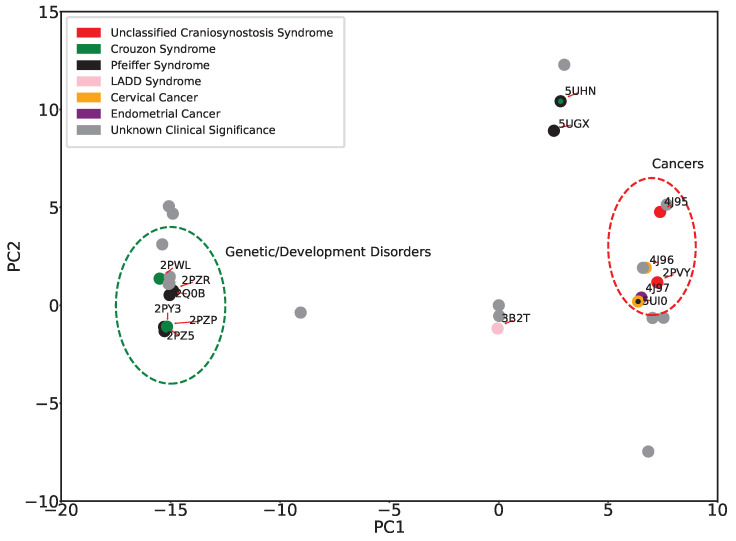
Eleven tertiary structures of FGFR2-TKD with known disease classifications are projected onto the top two PCs. The disease classifications consist of cervical and endometrial cancers, LADD syndrome, and unclassified cranial synostosis syndrome, in addition to Crouzon and Pfeiffer syndromes.

**Figure 8 ijms-25-04523-f008:**
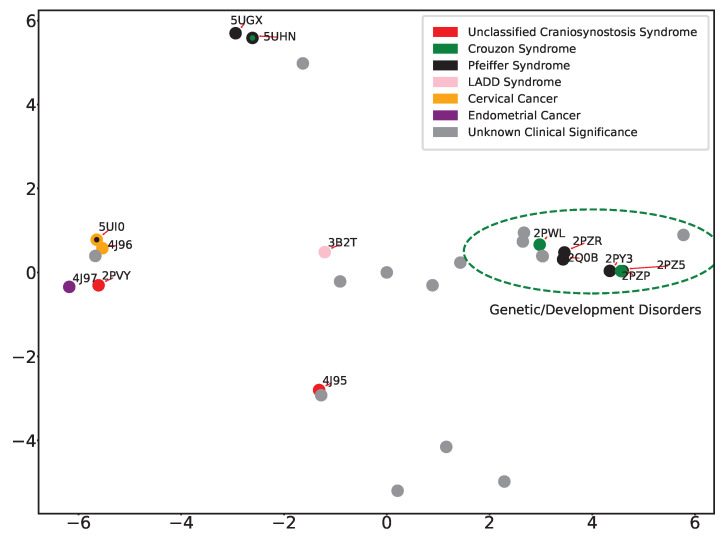
Eleven tertiary structures of FGFR2-TKD with known disease classifications are projected onto the top two Isomap components. The disease classifications consist of cervical and endometrial cancers, LADD syndrome, and unclassified cranial synostosis syndrome, in addition to Crouzon and Pfeiffer syndromes.

**Figure 9 ijms-25-04523-f009:**
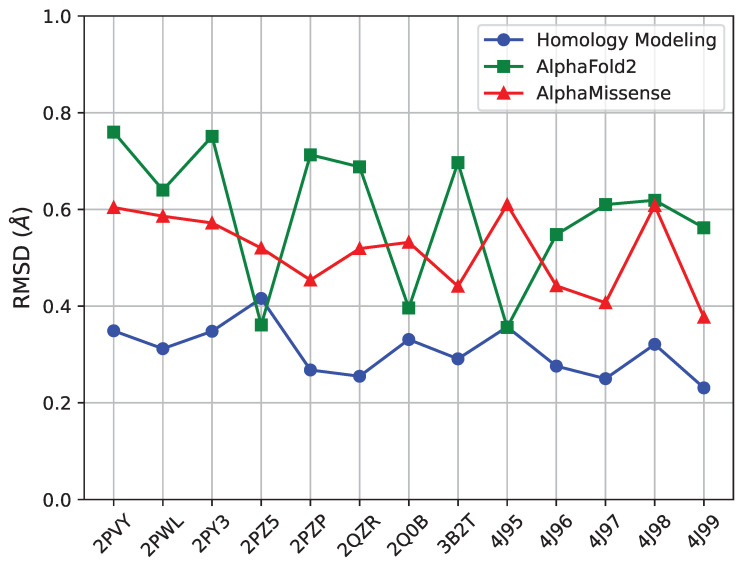
The RMSD between the known and predicted structure is shown here for each of the three methods (SWISS-MODEL, AlphaFold2, AlphaMissense) on 13 single-nucleotide variants. Compared to all methods, SWISS-MODEL has a lower RMSD per structure, no higher than 0.4 Å.

**Figure 10 ijms-25-04523-f010:**
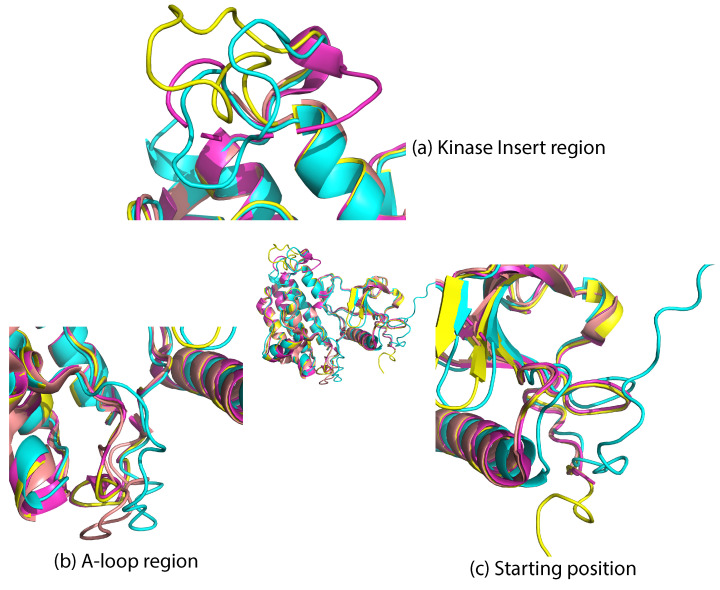
We zoom in here on the known tertiary structure with PDB ID 2PZ5 and its predicted structure from each of the three methods; pink color indicates the original structure 2PZ5, purple the SWISS-MODEL structure, yellow the AlphaMissense, and cyan the AlphaFold2 structure. The differences in modeling accuracy are made evident, particularly in three specific coil regions of the protein. (**a**) Insert Region: here, the models exhibit reduced effectiveness, which can be attributed to the complexity inherent in kinase insert regions, a lack of diverse data, and the challenges associated with integrating comprehensive bioactivity data. (**b**) Activation Loop: This section focuses on the coil areas of the protein, underscoring the limitations of AlphaFold2 and AlphaMissense in these dynamic regions. These limitations stem from biases in their training algorithms and their inability to adequately capture structural variations, especially when compared to the results of homology modeling. (**c**) Predictive Model Effectiveness: The effectiveness of predictive models is generally lower in the N-terminus region. This is due to various factors, including the inherent flexibility of these regions and so lack of agreement in training data.

**Figure 11 ijms-25-04523-f011:**
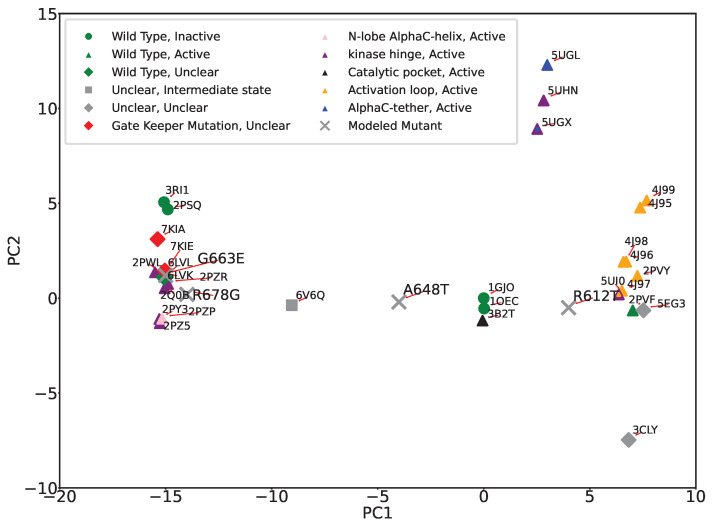
PC1-PC2 embedding of known and computed structures. Colors indicate functional regions, and shapes indicate state. ’X’ is reserved to indicate the computed structures.

**Figure 12 ijms-25-04523-f012:**
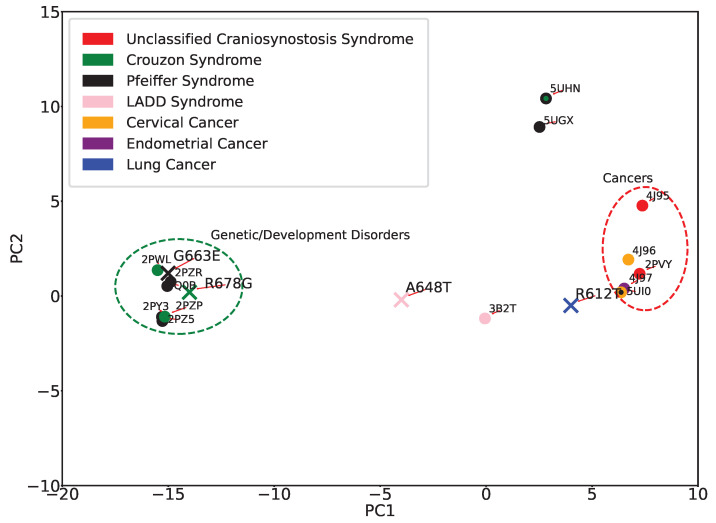
PC1-PC2 embedding of known and computed structures. Color-coding indicates disease classifications.

**Table 3 ijms-25-04523-t003:** The table presents data extracted from three databases: the Online Mendelian Inheritance in Man (OMIM), the Catalog of Somatic Mutations in Cancer (COSMIC), and the Human Gene Mutation Database (HGMD). It focuses on missense mutations located in the cytoplasmic regions of genes, linked to a range of disorders and cancers. These data enrich our original dataset from the PDB, which provides sequences with detailed structural information, now including disease associations. The table includes mutations from FGFR2 forms with no known structures, highlighted in red.

Cytoplasmic Region (399–821)			
Genetic/Development Disorders	Craniofacial Syndromes	Crouzon Syndrome	R678G
			2PWL, 2PZP
		Pfeiffer Syndrome	G663E
			2PY3, 2PZR
			2PZ5, 2Q0B
		UCS	2PVY, 4J95
	Syndromes with Multiple System Involvement	LADD1	3B2T
			A648T
Cancers	Gynecological Cancers	Cervical Cancer	4J96
		Endometrial Cancer	4J97
	Respiratory System Cancer	Lung Cancer	R612T

UCS: Unclassified Craniosynostosis Syndrome. LADD1: Lacrimo-auriculo-dento-digital Syndrome 1.

**Table 4 ijms-25-04523-t004:** This table summarizes the mutations in various FGFR2 structures, classified by their PDB ID or model, the specific mutation, the region name where the mutation occurs, the functional status of the protein (active, inactive, or unclear), and the associated disease.

PDB ID (Model)	Mutation Position	Region Name	Status	Disease
1GJO	WT	-	Inactive	-
1OEC	WT	-	Inactive	-
2PSQ	WT	-	Inactive	-
2PVF	WT	-	Active	-
3RI1	WT	-	Inactive	-
2PVY	K659N	Activation Loop	Active	UCS
2PWL	N549H	Kinase Insert	Active	CS
2PY3	E565G	Kinase Insert	Active	PS
2PZ5	N549T	Kinase Insert	Active	PS
2PZP	K526E	Alpha C helix at N-lobe	Active	CS
2PZR	K641R	Kinase Insert	Active	PS
2Q0B	E565A	Kinase Insert	Active	PS
3B2T	A628T	Catalytic Pocket	Active	LADD1
3CLY	C491A	-	Unclear	-
4J95	K659N	Activation Loop	Active	UCS
4J96	K659M	Activation Loop	Active	CC
4J97	K659E	Activation Loop	Active	EC
4J98	K659Q	Activation Loop	Active	-
4J99	K659T	Activation Loop	Active	-
5EG3	Multiple	Activation Loop	Active	-
5UGL	D650V	Alpha-C tether	Active	-
5UGX	E565A/D650V	Kinase Hinge/Alpha-C tether	Active	PS
5UHN	E565A/N549H	Kinase Hinge	Active	PS/CS
5UI0	E565A/K659M	Kinase Hinge/Activation Loop	Active	PS/CC
6LVK	-	WT	Unclear	-
6LVL	-	WT	Unclear	-
6V6Q	Multiple	-	Intermediate	-
7KIA	V564F	Gate Keeper	Unclear	-
7KIE	V564F	Gate Keeper	Unclear	-
Model	R612T	-	Unclear	LC
Model	A648T	Activation Loop	Unclear	LADD1
Model	G663E	Activation Loop	Unclear	PS
Model	R678G	-	Unclear	CS

UCS: Unclassified Craniosynotosis Syndrome. LADD1: Lacrimo-auriculo-dento-digital Syndrome 1. PS: Pfeiffer Syndrome. CC: Cervical Cancer. CS: Crouzon Syndrome. LC: Lung Cancer. EC: Endometrial Cancer.

**Table 5 ijms-25-04523-t005:** Sixteen single-nucleotide variants of FGFR2 are provided to AlphaMissense to obtain a computed pathogenicity. Column 4 shows whether a prediction is correct (1) or not (0).

Mutation Position	Pathogenic Score	Disease Type	Hit
K659N (2PVY, 4J95)	0.998 (likely pathogenic)	UCS	1
N549H (2PWL)	0.742 (likely pathogenic)	Crouzon Syndrome	1
E565G (2PY3)	0.982 (likely pathogenic)	Pfeiffer Syndrome	1
N549T (2PZ5)	0.805 (likely pathogenic)	Pfeiffer Syndrome	1
K526E (2PZP)	0.967 (likely pathogenic)	Crouzon Syndrome	1
K641R (2PZR)	0.828 (likely pathogenic)	Pfeiffer Syndrome	1
E565A (2Q0B)	0.982 (likely pathogenic)	Pfeiffer Syndrome	1
A628T (3B2T)	0.998 (likely pathogenic)	LADD1	1
K659M (4J96)	0.994 (likely pathogenic)	Cervical Cancer	1
K659E (4J97)	0.999 (likely pathogenic)	Endometrial Cancer	1
K659Q (4J98)	0.994 (likely pathogenic)	-	0
K659T (4J99)	0.996 (likely pathogenic)	-	0
R612T	0.97 (likely pathogenic)	Lung Cancer	1
A648T	0.976 (likely pathogenic)	LADD1	1
G663E	0.999 (likely pathogenic)	Pfeiffer Syndrome	1
R678G	0.971 (likely pathogenic)	Crouzon Syndrome	1

UCS: Unclassified Craniosynostosis Syndrome. LADD1: Lacrimo-auriculo-dento-digital Syndrome 1.

## Data Availability

Data is contained within the article.

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
