# Peer review of "Elucidating the Role of Wildtype and Variant FGFR2 Structural Dynamics in (Dys)Function and Disorder"

_ijms, 2024, doi:10.3390/ijms25084523_

Round 1
Reviewer 1 Report
Comments and Suggestions for Authors
In the paper entitled “Elucidating the Role of Wildtype and Variant FGFR2 Structural Dynamics in (Dys)Function and Disorder” the authors used PCA and isomap based analyses, to understand structural diversity present among PDB-deposited structures of wildtype and variant forms of FGFR2 and connected those observations towards different functional states of the protein. The interpretation of structural dynamics in terms of functionality in particular for a disease specific protein is definitely a prominent approach that has potential to benefit health industry. The overall presentation, clustering methods are very helpful to understand the message. However there are some queries that need to be clarified before publication of the article.
1. In introduction, the author initiated the possible dysregulation of RTK and corresponding diseases; then author introduced FGFR2 protein. However, the connection between RTK and FGFR2 is not clearly mentioned. Why did the author transit into FGFR2 from RTK, are they structurally or functionally similar? Are the potential diseases for dysregulation of those two different receptors correlated? How FGFR2 affects RTK signaling network?
2. Literature behind those two equations that assist in PCA analysis and its physical significance need to be addressed.
3. The algorithm or citation for sklearn.decomposition.PCA is required.
4. What could be the possible reason for PC1 not able to capture kinase hinge and nucleotide binding loop in Figure 4?
5. Why isomap is not as effective as PCA in clustering loop activation mutation; where lies the limitation in this method?
6. The pros and cons of other methods in addressing structural diversity of proteins in terms of pathogenicity need to be addressed briefly.
Author Response
We thank the reviewers for giving us the opportunity to improve our manuscript. We have addressed all their points one by one in the manuscript and in our answers below. We have also taken this opportunity to check our manuscript thoroughly for clarity and add text to further elaborate on some of the most important findings. All changes can be found in blue font in the revised manuscript.
Reviewer A
Reviewer comment 1: “In introduction, the author initiated the possible dysregulation of RTK and corresponding diseases; then author introduced FGFR2 protein. However, the connection between RTK and FGFR2 is not clearly mentioned. Why did the author transit into FGFR2 from RTK, are they structurally or functionally similar? Are the potential diseases for dysregulation of those two different receptors correlated? How FGFR2 affects RTK signaling network?”
Author response: We have revised our introduction to clarify the connection between receptor tyrosine kinases (RTKs) and FGFR2. We have highlighted that FGFR2 is a member of the RTK family, sharing structural and functional features with other RTKs. We have expanded on FGFR2's role in mediating cellular responses to fibroblast growth factors and its involvement in cell growth, survival, and differentiation. Additionally, we have clarified the link between FGFR2 dysregulation and various diseases, emphasizing its significance in the RTK signaling network and disease development.
Reviewer comment 2: “Literature behind those two equations that assist in PCA analysis, and its physical significance need to be addressed.”
Author response: We have expanded Section 2.2.1, "PCA for Characterization of Linear Structural Dynamics," to include a more comprehensive discussion of PCA's theoretical background and its practical applications in structural biology. We have specifically integrated references that underscore the importance of PCA in studying protein dynamics, the significance of root-mean-square deviation (RMSD) in structural comparisons, and the utility of PCA in exploring protein behavior in collective coordinate space.
Reviewer comment 3: “The algorithm or citation for sklearn.decomposition.PCA is required.”
Author response: Thank you for highlighting the need for a specific citation for the PCA algorithm implemented in sklearn.decomposition.PCA. Reference 41, which cites scikit-learn (Pedregosa et al., 2011)
Reviewer comment 4: “What could be the possible reason for PC1 not able to capture kinase hinge and nucleotide binding loop in Figure 4?”
Author response: We have addressed this issue in Section 3.1.2 in our manuscript. Several reasons contribute to this phenomenon, reflecting the inherent characteristics of PCA and the structural dynamics of the protein.
- Predominant Variance in Alternative Regions: The essence of PC1 is to encapsulate the direction along which the variance in the dataset is maximized. If significant structural changes are predominantly occurring in regions (activation loop) other than the kinase hinge and nucleotide binding loop, these areas will naturally be the focus of PC1.
- Limitations of Linear Analysis: Given that PCA operates on linear assumptions, it may not adequately capture the complexities of non-linear structural dynamics often exhibited by proteins. This limitation is particularly relevant if the kinase hinge and nucleotide binding loop engage in non-linear interactions or undergo non-linear conformational changes that PCA cannot effectively represent.
Reviewer comment 5: “Why Isomap is not as effective as PCA in clustering loop activation mutation; where lies the limitation in this method?”
Author response: The limitation of Isomap compared to PCA in clustering activation loop mutations in our analysis stems from Isomap's design to preserve geodesic distances, focusing on nonlinear dynamics. This characteristic, while valuable for capturing complex structural relationships, potentially reduces its effectiveness in clearly segregating linearly distinguishable variations, such as those from activation loop mutations. PCA, by emphasizing variance, more effectively “elevates”/clusters these mutations. However, as the manuscript states, isomap captures better than PCA.mutations in the gatekeeper region (in red) and the kinase hinge region (in purple)
Reviewer comment 6: “The pros and cons of other methods in addressing structural diversity of proteins in terms of pathogenicity need to be addressed briefly.”
Author response: We have incorporated a concise overview into our manuscript in the Discussion section in a new subsection numbered 4.1. This discussion aims to provide clarity on the selection of methods used in our study and their relative effectiveness in elucidating the relationship between structural variations and pathogenic outcomes in proteins. We thank the reviewer for prompting us to better organize the Discussion Section.
Reviewer 2 Report
Comments and Suggestions for Authors
The manuscript entitled “Elucidating the Role of Wild type and Variant FGFR2 Structural Dynamics in (Dys)Function and Disorder” was found interesting. I have critically review and found some points that need to be incorporated in minor revision.
1. In figure no.1, no captions are there, Add captions for each important area, region and different binding loop.
2. Table No.1, author should add residue length of the respective PDBID.
3. figure 10 should also have captions as per previous comment and author may reduce the legend.
4. Author should justify, why they chose 1GJO as wild type.
5. Author may add one schematic representation about whole story of research in introduction last part, this will create easy understanding to the readers.
6. Author can add more abbreviations.
7. Author can add separate conclusion part.
8. I did not find the Ramachandran plot for the models, Author can justify?

Author Response
We thank the reviewers for giving us the opportunity to improve our manuscript. We have addressed all their points one by one in the manuscript and in our answers below. We have also taken this opportunity to check our manuscript thoroughly for clarity and add text to further elaborate on some of the most important findings. All changes can be found in blue font in the revised manuscript.
Reviewer B
Reviewer comment 1: “In figure no.1, no captions are there, Add captions for each important area, region and different binding loop.”
Author response: We have captions for Figure 1. “Panel (a) illustrates the inactive state of FGFR2-TKD, indicating its readiness for activation through the display of the activation loop and key unphosphorylated tyrosine residues, distinguished by color coding for various functional regions. Panel (b) presents the FGFR2-TKD in its active, phosphorylated form, emphasizing phosphorylation's enhancement of substrate recognition and catalytic activity. The distinct positioning of the activation loop in both inactive and active states highlight its critical role in cellular growth and tissue repair processes.”
Reviewer comment 2: “Table No.1, author should add residue length of the respective PDBID.”
Author response: Thank you very much for your suggestion. We have added the residue length corresponding to each PDB ID in column 1 of Table 1.
Reviewer comment 3: “figure 10 should also have captions as per previous comment and author may reduce the legend.”
Author response: We apologize if somehow the captions for Figure 1 and Figure 10 were not visible to the reviewer. We did have captions for Figure 10. They started with: “We zoom here on the known tertiary structure with PDB ID 2PZ5 and its predicted structure from each of the three methods …”
Reviewer comment 4: “Author should justify, why they chose 1GJO as wild type.”
Author response: Yes, this is an important point on which we dwell in the manuscript. Indeed, we have deliberated on the suitability of various PDB IDs to represent the unphosphorylated FGFR2-TKD and ultimately selected 1GJO due to its appropriateness and superior resolution. Specifically, the decision was influenced by comparing potential candidates, including 1GJO, 1OEC, 2PSQ, and 3RI1, for their representativeness of an unphosphorylated FGFR2-TKD. The selection process involved assessing each structure's characteristics, with 1GJO emerging as the most suitable choice because of its higher resolution and the structural integrity it presents, making it a reliable reference for the unphosphorylated state of FGFR2-TKD. This discussion can be found detailed around lines 130-145 in the manuscript (section 2.1), where the selection rationale is explained, highlighting the crystal structure analysis and the importance of resolution in choosing 1GJO as the reference trace for subsequent analyses​​.
Reviewer comment 5: “Author may add one schematic representation about whole story of research in introduction last part, this will create easy understanding to the readers.”
Reviewer comment 6: “Author can add more abbreviations.”
Author response: We have done so sparingly, as we often have the experience of manuscripts becoming dense with abbreviations and becoming somewhat unreadable.
Reviewer comment 7: “Author can add separate conclusion part.”
Author response: Since the journal provides us with some freedom on this, we preferred to have a more detailed Discussion section that encapsulates conclusions as well.
Reviewer comment 8: “I did not find the Ramachandran plot for the models, Author can justify?”
Author response: Ramachandran plots inform us on the categorization of dihedral angles per secondary structure patterns/elements. The method operates over Cartesian coordinates, following research that shows that angles constitute cyclic coordinates presenting challenges for dimensionality reduction techniques. We follow literature convention on employing Cartesian coordinates.